# Mycobacterial and Human Ferrous Nitrobindins: Spectroscopic and Reactivity Properties

**DOI:** 10.3390/ijms22041674

**Published:** 2021-02-07

**Authors:** Giovanna De Simone, Alessandra di Masi, Alessandra Pesce, Martino Bolognesi, Chiara Ciaccio, Lorenzo Tognaccini, Giulietta Smulevich, Stefania Abbruzzetti, Cristiano Viappiani, Stefano Bruno, Sara Della Monaca, Donatella Pietraforte, Paola Fattibene, Massimo Coletta, Paolo Ascenzi

**Affiliations:** 1Dipartimento di Scienze, Università degli Studi Roma Tre, 00146 Roma, Italy; giovanna.desimone@uniroma3.it (G.D.S.); alessandra.dimasi@uniroma3.it (A.d.M.); 2Dipartimento di Fisica, Università di Genova, 16146 Genova, Italy; pesce@fisica.unige.it; 3Dipartimento di Bioscienze, Università di Milano, 20133 Milano, Italy; martino.bolognesi@unimi.it; 4Centro di Ricerche Pediatriche “R.E. Invernizzi”, Università di Milano, 20133 Milano, Italy; 5Dipartimento di Scienze Cliniche e Medicina Traslazionale, Università di Roma “Tor Vergata”, 00133 Roma, Italy; chiara.ciaccio@uniroma2.it; 6Dipartimento di Chimica “Ugo Schiff”, Università di Firenze, 50019 Sesto Fiorentino, Italy; lorenzo.tognaccini@unifi.it (L.T.); giulietta.smulevich@unifi.it (G.S.); 7Dipartimento di Scienze Matematiche, Fisiche e Informatiche, Università di Parma, 43124 Parma, Italy; stefania.abbruzzetti@unipr.it (S.A.); cristiano.viappiani@unipr.it (C.V.); 8Dipartimento di Scienze degli Alimenti e del Farmaco, Università di Parma, 43124 Parma, Italy; stefano.bruno@unipr.it; 9Servizio Grandi Strumentazioni e Core Facilities, Istituto Superiore di Sanità, 00161 Rome, Italy; sara.dellamonaca@iss.it (S.D.M.); donatella.pietraforte@iss.it (D.P.); paola.fattibene@iss.it (P.F.); 10Laboratorio Interdipartimentale di Microscopia Elettronica, Università Roma Tre, 00146 Roma, Italy

**Keywords:** heme, *Homo sapiens*, *Mycobacterium tuberculosis*, nitrobindin, structure, reactivity

## Abstract

Structural and functional properties of ferrous *Mycobacterium tuberculosis* (*Mt*-Nb) and human (*Hs*-Nb) nitrobindins (Nbs) were investigated. At pH 7.0 and 25.0 °C, the unliganded Fe(II) species is penta-coordinated and unlike most other hemoproteins no pH-dependence of its coordination was detected over the pH range between 2.2 and 7.0. Further, despite a very open distal side of the heme pocket (as also indicated by the vanishingly small geminate recombination of CO for both Nbs), which exposes the heme pocket to the bulk solvent, their reactivity toward ligands, such as CO and NO, is significantly slower than in most hemoproteins, envisaging either a proximal barrier for ligand binding and/or crowding of H_2_O molecules in the distal side of the heme pocket which impairs ligand binding to the heme Fe-atom. On the other hand, liganded species display already at pH 7.0 and 25 °C a severe weakening (in the case of CO) and a cleavage (in the case of NO) of the proximal Fe-His bond, suggesting that the ligand-linked movement of the Fe(II) atom onto the heme plane brings about a marked lengthening of the proximal Fe-imidazole bond, eventually leading to its rupture. This structural evidence is accompanied by a marked enhancement of both ligands dissociation rate constants. As a whole, these data clearly indicate that structural–functional relationships in Nbs strongly differ from what observed in mammalian and truncated hemoproteins, suggesting that Nbs play a functional role clearly distinct from other eukaryotic and prokaryotic hemoproteins.

## 1. Introduction

Globins are a superfamily of evolutionary conserved heme-proteins that bind, sense, and transport diatomic gases [1,2,3,4,5,6,7,8,9]. Most of these proteins (e.g., hemoglobin (Hb) and myoglobin (Mb)) are characterized by eight α-helical segments shaped around the heme with a 3/3 fold [1,7]. Besides these “canonical” all-α-helical globins, a smaller group of hemoproteins, named “truncated Hbs”, is characterized by a 2/2-fold [10,11]. In all-α-helical globins, the heme is placed deep within the protein cavity and takes contact with hydrophobic residues that prevent the oxidation of the heme-Fe(II) atom [1,3,4,7,9,11]. The side chain of the proximal HisF8 residue represents the fifth coordination ligand of the heme-Fe atom [1,3,4,9], whereas the E7 residue (mostly His and Tyr) represents the heme-Fe distal ligand that contributes to the modulation of iron reactivity and to the stability of the heme-bound ligand [1,3,4,9,12,13,14,15,16,17].

Over the last two decades, monomeric all-β-barrel and mixed-α/β hemoproteins, displaying globin-like reactivity, have been characterized. They include nitrophorins (NPs) (e.g., from *Rhodnius prolixus* and *Cimex lectularius*) and nitrobindins (Nbs) that have been found from bacteria to *Homo sapiens* [18,19,20,21,22,23,24,25,26]. In both NPs and Nbs, the penta-coordinated heme-Fe atom, which is highly exposed to the bulk solvent, is anchored to the protein by the proximal His residue [19,20,23,26,27,28,29,30].

Nbs display an anti-parallel β-barrel fold composed of 10 strands; protein loops contribute to the stabilization of the bound porphyrin ring, even though the heme-Fe atom is exposed to the solvent and is stably in the ferric form. While *Mycobacterium tuberculosis* Nb (*Mt*-Nb) and *Arabidopsis thaliana* Nb (*At*-Nb) are single-domain proteins, *Homo sapiens* Nb (*Hs*-Nb) is the *C*-terminal domain of the THAP4 protein [24,25,26,29,30,31].

Despite their negative redox potential, both *Mt*-Nb and *Hs*-Nb can be reduced under anaerobic conditions, allowing the detailed investigation of their spectroscopic and functional properties. This paper reports a deep investigation of spectroscopic and functional properties of ferrous *Mt*-Nb and *Hs*-Nb (i.e., *Mt*-Nb(II) and *Hs*-Nb(II), respectively). For the first time: (i) UV-Vis, RR, and EPR spectroscopic properties of ligand-free and ligand-bound Mt-Nb(II) and Hs-Nb(II) are recorded; (ii) kinetics of CO, NO and O_2_ binding to Mt-Nb(II) and Hs-Nb(II) was investigated by rapid-mixing stopped-flow technique and laser-flash photolysis; and (iii) spectroscopic and functional data were analyzed in parallel with those of *At*-Nb(II) [29]. Present data indicate that, upon ligand binding, the Fe(II) atom of Nbs moves onto the heme plane, this brings about a marked lengthening of the proximal Fe-imidazole bond, eventually leading to its rupture. This structural evidence is accompanied by a marked enhancement of ligand dissociation rate constants. Moreover, these data highlight the conservation of Nbs in bacteria, plants and animals, and indicate that structural–functional relationships in Nbs strongly differ from those of prototypical mammalian myoglobins, such as *Equus caballus* (*Ec*-Mb) and/or *Physeter catodon* (*Pc*-Mb), and of *Rhodnius prolixus* nitrophorins (*Rp*-NPs). This suggests that Nbs play a functional role clearly distinct from other eukaryotic and prokaryotic heme-proteins.

## 2. Results

### 2.1. UV-Vis and RR Spectroscopic Properties of Mt-Nb(II) and Hs-Nb(II)

The UV-Vis spectra of unliganded *Mt*-Nb(II) and *Hs*-Nb(II) are very similar and show the typical pattern of a pure five-coordinate high spin species (Figure 1A), as confirmed by the high-frequency RR spectra (Figure 1B), in which the core-size marker bands were observed at 1472–1473 (ν_3_), 1558–1568 (ν_2_), and 1606 cm^−1^ (ν_10_). The (C = C) vinyl stretching modes give rise to the band at 1621 cm^−1^ in *Mt*-Nb(II), whereas two bands were observed at 1621 and 1627 cm^−1^ in *Hs*-Nb(II) suggesting a different orientation of the two vinyl groups, as observed in *Hs*-Mb(III) (26). The corresponding bending modes were observed at 417 cm^−1^ (Figure 1B). The δ(C_β_C_c_C_d_) propionate in-plane bending mode gives rise to a band at 365 cm^−1^ with a shoulder at 377 cm^−1^ in *Mt*-Nb(II), and a broad band at 371 cm^−1^ in *Hs*-Nb(II).

In the low-frequency region of the spectrum, obtained with the 441.6 nm excitation line, an intense band was observed at 213 cm^−1^ that decreases upon excitation with the 413.1 nm in both *Mt*-Nb(II) and *Hs*-Nb(II) (Figure 1B). This band was assigned to the ν(Fe-His) stretching mode since it is expected to give rise to a strong band in five-coordinate high spin ferrous heme proteins upon excitation in the Soret band [32]. The ν(Fe-His) stretching mode frequency spans from about 200 cm^−1^ (neutral proximal His) to 250 cm^−1^ (deprotonation of N_δ_, as in the heme-containing peroxidases). Its frequency, very sensitive to the protein matrix, is an optimum probe of the proximal cavity structure [33]. In ferrous *Ec*-Mb(II) and *Pc*-Mb(II), where the N_δ_ proton is H-bonded to a neutral backbone carbonyl group, the ν(Fe-His) stretch was found at 220 cm^−1^ [34,35]. Likewise, in the Nbs(II), where the N_δ_ proton is H-bonded to a neutral backbone ND1 of Lys26 (*Mt*-Nb) or with the O atom of Thr29 (*Hs*-Nb), the frequency of the ν(Fe-His) band is at 213 cm^‒1^, indicating a weaker interaction than in Mb (Figure 1B).

### 2.2. UV-Vis and RR Spectroscopic Properties of Mt-Nb(II)-CO and Hs-Nb(II)-CO

CO was found very informative to examine the distal cavity of heme-proteins [36]. In fact, the back-donation from the Fe dπ to the CO π* orbitals depends on polar interactions. A very important role is played by H-bonds between the bound CO and the distal protein residues. A strong H-bond favors back-donation, with a strengthening of the Fe-C bond and a correspondingly weakening of the CO bond [37].

Within this context, a linear correlation with a negative slope between the frequencies of the ν(Fe-C) and ν(CO) stretching modes was found for a large class of carbonylated heme-proteins and heme-model compounds containing imidazole as the fifth heme-Fe(II) ligand (Figure 1C) [38]. The ν(Fe-C)/ν(CO) position along the correlation line reflects the type and strength of distal polar interactions [36]. Wild type *Ec*-Mb(II)-CO and *Pc*-Mb(II)-CO is characterized by moderate back-bonding induced by weak H-bonding from the distal His residue (*Ec*-Mb(II)-CO: 509 and 1944 cm^−1^; *Pc*-Mb(II)-CO: 508 and 1944 cm^−1^). When the distal His residue is replaced by non-polar residues (e.g., in the His64Val mutant of the *Pc*-Mb), the ν(Fe-C)/ν(CO) point slides down the line (488 and 1966 cm^−1^), reflecting the expected decrease in back-bonding [37]. Variations in the donor strength of the trans-ligand also affect the frequencies. In fact, CO complexes with a weak or absent proximal ligand are located above the histidine line [36]. Hence, the upper dashed line in Figure 1C represents either five-coordinate heme-Fe(II)-CO complexes with no trans-ligand or six-coordinate heme-Fe(II)-CO adducts with weak trans-ligands [36,39].

Upon CO binding, *Mt*-Nb(II) and *Hs*-Nb(II) give rise to a six-coordinate low spin complex between pH 6.0 and 10.2 with UV-Vis absorption bands at 419, 537, and 568 nm (Figure 1A). The RR modes of the *Mt*-Nb(II)-CO complex were identified by an isotopic shift at 510 cm^−1^ ν(Fe-C) and 1958 cm^−1^ ν(CO) (Figure 1C). This latter value is very close to that obtained for the *At*-Nb(Fe(II)-CO complex by FTIR [29]. The RR modes of *Hs*-Nb(II)-CO show a similar ν(Fe-C) mode at 509 cm^−1^, but the intense fluorescence in the 1900 to 2000 cm^−1^ region observed in the *Hs*-Nb(II)-^13^CO sample does not allow us to identify the ν(^13^CO) mode. The ν(CO) mode was tentatively assigned to the band observed at 1950 cm^−1^ (Figure 1C). The ν(Fe-C)/ν(CO) position for both Nbs appears displaced above the solid His line, the effect being more pronounced for *Mt*-Nb(II)-CO than for *Hs*-Nb(II)-CO, moving toward frequencies typical of CO complexes with no or weakly-bound trans-ligand (Figure 1C). Therefore, this behavior might reflect a weaker proximal Fe-His bond in the two carbonylated Nb(II) with respect to mammalian Mbs.

### 2.3. UV-Vis and EPR Spectroscopic Properties of Mt-Nb(II)-NO and Hs-Nb(II)-NO

Extensive studies support the view that the UV-Vis and EPR spectroscopy of ferrous nitrosylated heme-proteins and heme-model compounds are indicative of the strength of the proximal His-Fe(II) bond and in turn of the ferrous metal center reactivity [1,40,41,42,43,44,45,46,47] 

Absorption spectra of the Fe(II)-NO derivative of *Ec*-Mb, *Mt*-Nb, and *Hs*-Nb are reported in Figure 2A. The difference between these heme-proteins is strikingly remarkable with a blue-shift of *Mt*-Nb(II)-NO and *Hs*-Nb(II)-NO, associated to a marked decrease of the extinction coefficient, both features suggesting a weakening of the heme-Fe-His proximal bond [48].

Mirroring what already observed for absorption spectra of Figure 2A, a dramatic difference between the nitrosylated hemoproteins clearly comes out by the EPR spectra performed at 110 K (Figure 2B–D). Thus, heme proteins in a histidine–Fe(II)–NO conformation are characterized by a temperature-dependent EPR spectrum composed of a combination of two paramagnetic species whose relative composition depends on the temperature, such that (a) at high temperatures (>150 K) the cw-EPR spectra are dominated by an axial species (denoted state A, from axial) (called *SysA* below), and (b) at low temperature (< 150 K) a rhombic species prevails (species R, from rhombic) (called *SysR* below). When the histidine-iron bond is elongated or broken, the effect of the histidine nitrogen on the EPR spectrum is lost. Consequently, the EPR spectrum is only split by the NO nitrogen and resolved into three sharp lines with a hyperfine splitting constant of 17 G (called *Sys5C* below) [1,40,41,42,43,44,45,46,47].

In Figure 2B–D, the experimental EPR spectra are compared with their simulations, employing different percentages of the three forms (i.e., *SysA*, *SysR* and *Sys5C*), which are reported in the figure caption. On the basis of this simulation, the EPR signal of *Ec*-Mb(II)-NO (Figure 2B), detected at pH 7.0, displays a full (~100%) rhombic shape with some resolution of the superhyperfine structure in the *g*_z_ region of the spectrum characteristic of a hexa-coordinated form [49]. Conversely, in the case of *Hs*-Nb(II)-NO and *Mt*-Nb(II)-NO, indeed a three-line pattern in the high magnetic field region of EPR spectra was detected at pH 7.0 (Figure 2C,D), even though important spectroscopic differences occur between the two Nb(II)-NO. Thus, in the case of *Hs*-Nb(II)-NO we observe at pH 7.0 the predominance of the species *Sys5C* (~88%), characterized by the three-line hyperfine structure (Figure 2C), clearly indicating that most molecules display the five-coordination of the heme-Fe(II)-NO species as the result of cleavage of the proximal His-Fe bond. On the other hand, the EPR spectrum of *Mt*-Nb(II)-NO exhibits a mixture of different forms, with only 55% attributable to the species *Sys5C*, as from simulations (Figure 2D). Therefore, like for CO-bound (see above), and even to a higher extent, in NO-bound Nb(II) the evidence for a weak proximal bond emerges in a clear cut fashion, suggesting that upon distal ligand binding a dramatic strain is exerted on the proximal Fe-His bond, eventually leading to the cleavage of the Fe-His proximal bond in a large percentage of molecules. 

### 2.4. Kinetics of CO Binding to Mt-Nb(II) and Hs-Nb(II)

#### 2.4.1. Rapid-Mixing

The time course of CO binding to *Mt*-Nb(II) and *Hs*-Nb(II) by rapid-mixing technique is strictly monophasic (>95%) (Figure 3A,B) and wavelength-independent. The amplitude of the exponentials is dependent on the CO concentration under all the experimental conditions, since CO concentration is similar to the value of the dissociation equilibrium constant K_(CO)_ (i.e., [CO] does not fully saturate *Mt*-Nb(II) and *Hs*-Nb(II)) (Figure 3A,B). Values of *k*_obs(CO)_ are independent of the heme-protein concentration (Appendix A) and increase linearly with the CO concentration over the whole CO concentration range explored (between 2.0 × 10^‒5^ M and 2.0 × 10^‒4^ M). In Figure 3A,B are reported some kinetic progress curves for CO binding to *Hs*-Nb(II) (Figure 3A) and *Mt*-Nb(II) (Figure 3B); the analysis of data, shown in Figure 3C according to Equation (2), allowed to determine values of *k*_on(CO)_ and *k*_off(CO)_ for (de)carbonylation of *Mt*-Nb(II)(-CO) and *Hs*-Nb(II)(-CO). Moreover, values of *k*_off(CO)_ for decarbonylation of *Mt*-Nb(II)-CO and *Hs*-Nb(II)-CO were obtained by CO displacement with NO (Figure 4). The time course of CO displacement from *Mt*-Nb(II)-CO and *Hs*-Nb(II)-CO by NO, investigated by rapid-mixing technique, is strictly monophasic (>93%), as indicated by the distribution of residuals (Figure 4).

Values of *k*_on(CO)_ for CO binding to *Mt*-Nb(II) and *Hs*-Nb(II) are 2- to 4-fold slower, respectively, than that reported for *At*-Nb(II) carbonylation (=2.3 × 10^5^ M^−1^ s^−1^) [29], and even slower (i.e., 5- and 10-folds, respectively) than that of *Pc*-Mb(II) [56] (Table 1). The unusually high values of *k*_off(CO)_ for CO dissociation from *Mt*-Nb(II)-CO and *Hs*-Nb(II)-CO, as derived from linear plots of *k*_obs(CO)_ versus [CO] (Figure 3C), are closely similar with those directly measured following CO displacement by NO (Figure 4). The values of *k*_off(CO)_ are ~140-fold higher than that reported for *At*-Nb(II)-CO decarbonylation [29] and ~500-fold higher than those observed in mammalian Mbs (e.g., *Pc*-Mb(II)) [15]. The resulting values of the dissociation equilibrium constant for CO binding to *Mt*-Nb(II) and *Hs*-Nb(II) (i.e., *k*_off(CO)_/*k*_on(CO)_) are very high, being 6.3 × 10^‒5^ M and 3.8 × 10^−5^ M, respectively. These values are about 200-fold and 1000-fold higher than those of *At*-Nb(II) (2.2 × 10^−7^ M) [29] and mammalian Mbs (e.g., *Ec*-Mb(II), 5.7 × 10^−8^ M; and *Pc*-Mb(II), 3.7 × 10^−8^ M) [15,52].

The low reactivity of CO for *Mt*-Nb(II) and *Hs*-Nb(II) (Figure 3), as compared to that for *Pc*-Mb(II) [15] (Table 1), may be ascribed either to (i) proximal effects, possibly related to a higher activation free energy for the in-plane motion of the Fe-His proximal bond [56] and/or to (ii) distal effects, due to either the steric hindrance exerted by the heme distal residues (His85 in *Mt*-Nb and Thr91 in *Hs*-Nb), altering the Fe(II)-C-O angle [57] and/or crowding of H_2_O molecules in the vicinity of the heme because of the exposure of the distal side to the bulk solvent. The unusually high CO dissociation rate constant from *Mt*-Nb(II)-CO (Figure 4) indeed may reflect a weakening of the heme-Fe-His proximal bond for the carbonylated species, as suggested by the unusual ν(Fe-C)/ν(CO) position (Figure 1C). Such a feature was also observed in other hemoproteins, such as soluble guanylate cyclase, cytochrome *c*, and sensor proteins (e.g., FixL) [58,59], accompanied by an increase of the CO dissociation rate constant, as observed for heme model compounds [60].

An additional piece of information may come from the pH dependence of CO binding to both *Hs*-Nb and *Mt*-Nb, which does not show any enhancement over the 2.2–7.0 pH range (Figure 5A). This behavior is drastically different from what observed in most of the other hemoproteins, wherefore at pH < 5.0 a relevant increase of the CO binding rate constant is observed with variable p*K*_a_ values [56,61,62,63,64,65,66,67]. This feature, which was attributed to the cleavage (or severe weakening) of the heme-Fe-His proximal bond in the unliganded form, as demonstrated by the spectroscopic features, is characterized by the blue-shift of the absorption spectrum in the Soret region and the appearance of two peaks at 525 and 565 nm [56,61,62]. However, in the case of Nbs the pH independence of the CO binding rate constants (Figure 5A) is mirrored by an absorption spectrum of the deoxygenated form which remains unchanged for 1 s (keeping the same features as at pH 7.0) even at pH 2.2 (Figure 5B) before decaying for denaturation (data not shown). This clearly indicates that the heme-Fe-His proximal bond remains unaltered even at this low pH value, likely reflecting a highly compact proximal side of the heme pocket in the unliganded form of *Hs*-Nb(II) and *Mt*-Nb(II), which dramatically lowers the p*K*_a_ of the proximal bond.

#### 2.4.2. Rebinding Kinetics

The progress curves of CO rebinding to *Mt*-Nb(II) and *Hs*-Nb(II) are characterized by spectral changes reflecting: (i) geminate CO rebinding, (ii), bimolecular carbonylation process and (iii) conformational changes.

The negligible or low CO geminate rebinding for *Mt*-Nb(II and *Hs*-Nb(II)), respectively, (Figure 6 and Figure 7) may either arise from the low heme reactivity or from the easy escape of the ligand from the distal pocket. It turns out that in both Nbs *φ*_gem_ ≤ 0.05, indeed suggesting that *k*_BC(CO)_ >> *k*_BA(CO)_ (Equation (3)); in this respect, this outcome is consistent with the structural evidence of a remarkable heme exposure to the solvent and the absence of relevant distal structural constraints, thus leading to a very low energy barrier for the escape to the solvent of the photolyzed CO.

Most of the absorption change for CO rebinding kinetics to *Mt*-Nb(II) (Figure 6A) is due to bimolecular rebinding, which is best described by a sum of two exponential decay functions, even though the two kinetic phases display similar amplitudes and only a two-fold difference for observed rate constants (Figure 6B). Interestingly, values of the bimolecular rates constants at pH 7.4 and 20.0 °C (i.e., *k*^1^_on(CO)_ = (1.6 ± 0.01) × 10^5^ M^‒1^ s^‒1^ and *k*^2^_on(CO)_ = (0.83 ± 0.01) × 10^5^ M^‒1^ s^‒1^) are only slightly (2- to 3-fold) faster than what observed by stopped-flow (Table 1). This difference, though quite small, indeed might suggest that immediately after the CO detachment the heme is in a somewhat faster conformation, relaxing over the ms time regime (that is a time interval overlapping with CO rebinding) to a structural arrangement characterized by a slightly higher (by ~3 kJ/mole) energy barrier for CO binding. This occurrence seems supported by the evidence (Figure 6A) that the progress curves of CO rebinding to *Mt*-Nb(II) show a small increase in the signal over the microsecond time scale, which is likely due to a protein conformational change following the photodissociation of the bound ligand [50]. This signal is independent of the CO concentration and is weakly temperature-dependent. To highlight this signal, we collected the absorbance change at 421.5 nm, which is an isosbestic point of the spectral difference between carbonmonoxy- and deoxy- species. This signal (on a × 4 scale) is compared with the one measured at 436 nm in Figure 6C. The time course shows the typical shape for a time-extended conformational change observed in many heme-proteins [68] and it can be described with a stretched exponential decay with the time constant of 770 μs and a stretching exponent of 0.24, followed by exponential relaxation with a lifetime identical to the long-lived decay detected at 436 nm, corresponding to CO rebinding. Therefore, the overlapping of the heme relaxation time with CO rebinding time is responsible for the non-exponential behavior of the conformational transition and the apparent multiple exponential behavior of the CO rebinding, which occurs with a continuum of reactivity-changing species. Figure 6B shows that the amplitude of the slow decay systematically increases at lower CO concentrations, a fact that is expected for such transitions. Moreover, the slower bimolecular rate, observed in flash photolysis, is similar to the one measured in the stopped-flow experiments, which may be taken as a further hint towards the identification of the slow phase as a relaxed deoxy structure which is functionally distinct from the liganded state [69].

Somewhat different behavior is observed for CO recombination to *Hs*-Nb(II), wherefore appreciable geminate recombination is detected (amounting to ~ 5 % of the total amplitude of the rebinding process) with a *r_gem_* = 9 × 10^7^ s^−1^ at 25 °C. According to Equation (3) it suggests that for *Hs*-Nb(II) *k*_BA(CO)_ ≈ *r*_gem_ × *φ*_gem_ ≈ 4.5 × 10^6^ s^−1^ and *k*_BC(CO)_ ≈ *r*_gem_ -*k*_BA(CO)_ ≈ 8.5 × 10^7^ s^−1^. Furthermore, the bimolecular rebinding is more markedly biphasic in *Hs*-Nb(II) than in *Mt*-Nb(II) (Figure 6A and Figure 7A), mostly because of a minor faster phase (corresponding to ~ 15% of the absorption change due to the bimolecular process) with *k*^f^_on(CO)_ = (3.9 ± 0.1) × 10^6^ M^‒1^ s^‒1^ at 25.0 °C (Figure 7A). On the other hand, the second-order rate constant of the slower process (i.e., *k*^s^_on(CO)_ = (1.7 ± 0.1) × 10^5^ M^‒1^ s^‒1^, corresponding to ~ 85% of the absorption change due to the bimolecular process) is closely similar to what observed by stopped-flow (Table 1 and Figure 3A and Figure 7A). As reported for *Mt*-Nb(II), the CO rebinding kinetics shows a small rise in the micro-seconds timescale, possibly reflecting a conformational relaxation following photolysis. Unlike the case of *Mt*-Nb(II), it was not possible to identify a wavelength at which the structural relaxation could be clearly observed. However, the larger amplitude of the slow phase and its rate very close to that observed by stopped-flow indeed suggest that the relaxation is likely faster than in the case of *Mt*-Nb(II) and it gets closer to completion during the time scale of the flash photolysis experiment. However, the larger extent of geminate recombination phase and a faster bimolecular recombination process also indicates that in *Hs*-Nb(II) after CO detachment the heme is in a higher reactivity structural arrangement, as indicated by the 7 kJ/mole lower free energy barrier than in *Mt*-Nb(II) for CO binding to the faster process (Table 2).

From the temperature dependence of the bimolecular rebinding rate linear Eyring plots of *k*_on(CO)_ can be obtained between 10.0 °C and 40.0 °C, allowing to determine for both Nbs values of the activation enthalpy and entropy (Table 2). The amplitude of each phase was not influenced by the temperature (Figure 6B and Figure 7B), while the CO concentration seemed to have a small systematic effect consistent with the hypothesis that the slow phase is populated after a structural relaxation. Interestingly, for the slower bimolecular CO rebinding process all activation parameters (i.e., Δ*G*_2_^‡^, Δ*S*_2_^‡^ and Δ*H*_2_^‡^, Table 2) are closely similar between *Hs*-Nb(II) and *Mt*-Nb(II), clearly indicating that their structural arrangement is essentially the same after the conformational change following the CO detachment. On the other hand, a striking difference can be observed between the two Nbs before this structural transition (Table 2); thus, in the faster process, observed in *Hs*-Nb(II), the lower free energy barrier is fully attributable to a much lower activation entropy, which is essentially 0, as compared to the very negative value observed in *Mt*-Nb(II) (Table 2).

### 2.5. NO Binding to Mt-Nb(II) and Hs-Nb(II)

#### 2.5.1. Rapid Mixing

The time course of *Mt*-Nb(II) and *Hs*-Nb(II) nitrosylation is strictly monophasic (>91%) (Figure 8A,B) and wavelength-independent. The amplitude of the exponentials is independent of the NO concentration under all the experimental conditions since the NO concentration was larger by at least three orders of magnitude than the dissociation equilibrium constant *K*_(NO)_ (i.e., [NO] was largely sufficient to saturate both *Mt*-Nb(II) and *Hs*-Nb(II)) (Figure 8A,B). The values of *k*_obs(NO)_ are independent of the heme-protein concentration and increase linearly with the NO concentration over the whole gaseous ligand concentration range explored (between 1.5 × 10^‒5^ M and 1.0 × 10^‒4^ M). The analysis of data, shown in Figure 8C according to Equation (4), allowed to determine only values of *k*_on(NO)_ for the nitrosylation of *Mt*-Nb(II) and *Hs*-Nb(II) (1.7 × 10^6^ M^−1^ s^−1^ and 9.3 × 10^5^ M^−1^ s^−1^, respectively). In fact, the intercept of the straight lines with the *y* axis is close to zero. Therefore, the values of *k*_off(NO)_ for *Mt*-Nb(II)-NO and *Hs*-Nb(II)-NO denitrosylation (6.8 × 10^−2^ s^−1^ and 2.1 × 10^−2^ s^−1^, respectively) were obtained by NO displacement with CO. The time course of NO displacement from *Mt*-Nb(II)-NO and *Hs*-Nb(II)-NO by CO (i.e., of *Mt*-Nb(II)-CO and *Hs*-Nb(II)-CO formation) is strictly monophasic (>96%) (Figure 9). 

Values of *k*_on(NO)_ for NO binding to *Mt*-Nb(II) and *Hs*-Nb(II) (1.7 × 10^6^ M^−1^ s^−1^ and 9.3 × 10^5^ M^−1^ s^−1^, respectively) are 50- to 100-fold slower, respectively, than that reported for *At*-Nb(II) nitrosylation (=8.1 × 10^7^ M^−1^ s^−1^) [29], and 10- and 20-folds, respectively, of that observed for *Pc*-Mb(II) (2.2 × 10^7^ M^−1^ s^−1^) [53] (Table 1). The unusually high values of *k*_off(NO)_ for NO dissociation from *Mt*-Nb(II)-NO and *Hs*-Nb(II)-NO (6.8 × 10^−2^ s^−1^ and 2.1 × 10^−2^ s^−1^, respectively) (Figure 9) are similar to that for NO dissociation from *At*-Nb(II)-NO (~8 × 10^−2^ s^−1^) [29] (Table 1). The values of *k*_off(NO)_ for the denitrosylation of *Mt*-Nb(II)-NO, and *Hs*-Nb(II)-NO, as well as for *At*-Nb(II)-NO [29], are 200- to 800-fold faster than those reported for the denitrosylation of mammalian Mbs (e.g., *Pc*-Mb(II); *k*_on(NO)_ = 1.2 × 10^−4^ s^−1^) [54] (Table 1). Lastly, the affinity of NO (i.e., *K* = *k*_off(NO)_/*k*_on(NO)_) for the fast-reacting form of *Hs*-Nb(II) (=1.4 × 10^−9^ M), calculated using the *k*_on(NO)_ value determined by laser photolysis, is similar to that of *At*-Nb(II) (~1 × 10^−9^ M), obtained with the same approach [29]. On the other hand, the NO affinity for the slow-reacting form of *Hs*-Nb(II) (=2.5 × 10^−8^ M) calculated using the *k*_on(NO)_ value determined by laser photolysis agrees with those of *Mt*-Nb(II) and *Hs*-Nb(II) (i.e., *k*_off(NO)_/*k*_on(NO)_ = 4.0 × 10^−8^ M and 2.3 × 10^−8^ M, respectively) calculated with *k*_on(NO)_ values determined by rapid mixing technique (Table 1). Of note, the affinity of NO for Nbs(II) is lower than that of mammalian Mb(II), displaying values of *k*_off(NO)_/*k*_on(NO)_ for Nb(II) nitrosylation which are 200- to 10,000-fold higher than that of *Pc*-Mb(II) (=5.5 × 10^−12^ M) [53,54] (Table 1).

#### 2.5.2. Rebinding Kinetics

After nanosecond laser photolysis of NO, the time course of *Hs*-Nb(II) nitrosylation displays: (i) a geminate rebinding phase, corresponding to about 35% of the total recombination absorption change (φ_gem_ = 0.35) with an apparent lifetime of 10 ns (*r_gem_* ≈ 6.9 × 10^7^ s^−1^, Equation (3), and (ii) a bimolecular biphasic phase, characterized by a faster process (~8% of the total recombination absorption change with *k*_on(NO)_ = 1.5 × 10^7^ M^−1^s^−1^) and a slower one (~57% of the total recombination absorption change with *k*_on(NO)_ = 8.5 × 10^5^ M^−1^s^−1^) (Figure 10). The higher geminate recombination underlies a quite fast recombination rate *k*_BA(NO)_ (≈ 2.3 × 10^7^ s^−1^), about 6 times faster than that for CO, while the escape rate constant turns out to be fairly similar for CO and NO, reflecting the substantially similar size of the two ligands. The faster value of the second-order rate constant for NO binding is closely similar to what observed for *Pc*-Mb (Table 1 and [53]) and about 6-fold slower than that reported for *At*-Nb(II) [29]. On the other hand, the slower value of the second-order rate constant for NO binding to *Hs*-Nb(II) species is about 20-fold and 100-fold slower than what reported for *Pc*-Mb and *At*-Nb(II), respectively (Table 1).

### 2.6. O_2_ Dissociation from Mt-Nb(II)-O_2_ and Hs-Nb(II)-O_2_

As reported for heme-based sensors [59], the highly solvent-exposed heme-Fe(II) atom [26] of *Mt*-Nb(II) and *Hs*-Nb(II) undergoes instantaneous O_2_-mediated oxidation, the auto-oxidation rate being 10^4^–10^5^ times larger than that of *Pc*-Mb(II) [16]. Therefore, only values of the first-order rate constant for O_2_ dissociation from *Mt*-Nb(II)-O_2_ and *Hs*-Nb(II)-O_2_ species (i.e., *k*_off (O2)_) were determined by oxygen pulse experiments. The mono-exponential time courses of *Mt*-Nb(II)-O_2_ and *Hs*-Nb(II)-O_2_ deoxygenation are reported in Figure 11. *k*_off(O2)_ values of *Mt*-Nb(II)-O_2_ and *Hs*-Nb(II)-O_2_ (1.1 × 10^1^ s^−1^ and 1.9 × 10^1^ s^−1^, respectively) are similar to those of *At*-Nb(II)-O_2_ (6.8 s^−1^) and of *Pc*-Mb(II)-O_2_ (1.0 × 10^1^ s^−1^) [29,51].

## 3. Discussion 

The highly solvent exposed heme-Fe-atom is at the root of the fast auto-oxidation rate of *Mt*-Nb(II), *At*-Nb(II) and *Hs*-Nb(II), which is similar to that of *Rp*-NPs and 10^4–^10^5^ times higher than that of mammalian globins [22,29,30]. This impairs oxygenation, carbonylation and nitrosylation of Nb(II)s under non-reducing conditions. Nonetheless, the kinetic and thermodynamic behavior of Nb(II) forms can be investigated under appropriate conditions which prevent autoxidation to significantly affect the investigation.

Remarkably, the easy access to the heme pocket of Nbs [26,29,30] does not lead to fast ligand binding rate constants (Table 1 and Table 2), indicating that the easier access pathway does not affect to a meaningful extent the activation free energy of ligand binding to the heme-Fe(II) atom of Nbs. Actually, the much slower CO binding rate constants for *Hs*-Nb and *Mt*-Nb (Table 1), which display an activation free energy (Δ*G*^‡^ = 43.7 kJ/mol for *Hs*-Nb and 45.0 kJ/mol for *Mt*-Nb) much higher than that of *Ec*-Mb (Δ*G*^‡^ = 39.2 kJ/mol) and *Pc*-Mb (Δ*G*^‡^ = 39.7 kJ/mol), might stem from crowding of H_2_O molecules in the distal side of the heme pocket, as observed from X-ray structures of Fe(III) Nbs [26], which would raise the free energy barrier for ligand binding to the heme’s Fe atom. However, an additional contribution might arise from a higher energy barrier for the in-plane movement of the unliganded heme-Fe-His proximal bond to bind CO [56,61,62,63,64,65,66,67]. The strain, exerted on the proximal His-Fe(II) bond by this movement, may be due to the clustering of amino acid side chains in the proximal side of the heme pocket, which might be also responsible for the resistance of the proximal bond even at very low pH values (Figure 5). As a matter of fact, an inspection of the available protein three-dimensional structures of *Mt*-Nb(III) and *Hs*-Nb(III) supports this hypothesis [26], suggesting that the increased clustering of residues in the proximal Nb heme pocket, relative to Mb, can be related to the different surrounding secondary structures (β-strands in Nbs versus α-helices in Mbs), which imply different residue spacing and structural arrangement around the heme group. In particular, the *Mt*-Nb three-dimensional structure [26] shows that five amino acid residues (i.e., Ile30, Phe33, Tyr35, Met145, and Leu156) directly contact the porphyrin ring on the proximal side through van der Waals interactions (distance ≤ 4.0 Å). Such a scheme of contacts is also observed in *At*-Nb [29] and *Hs*-Nb [30]. On the other hand, on the proximal side of *Pc*-Mb [70] and *Ec*-Mb [71] only three amino acid residues (i.e., Leu89, His97, and Ile99) are in contact with the heme group. 

This strain, imposed by the protein structure, would also explain the very weak proximal His-Fe(II) bond in (i) the CO-bound form, as indicated by the resonance Raman frequencies of the ν(Fe-C) and ν(CO) modes (Figure 1C), and in (ii) the NO-bound form, as indicated by the absorption and EPR spectroscopy (Figure 2). This effect is then mirrored by the functional behavior of liganded forms, wherefore much faster CO dissociation rate constants (Figure 4 and Table 1) as well as faster NO dissociation rate constants (Figure 9 and Table 1) are observed, being in keeping with a severe weakening (or even a cleavage) of the proximal Fe-His bond in the liganded species. This peculiar structural arrangement of the liganded forms of both Nbs finds further support in the relatively slow relaxation (overlapping with the bimolecular recombination process) toward the equilibrium reformation of the Fe-His bond in the unliganded species, which brings about a multi-exponential rebinding both for CO (Figure 6 and Figure 7 and Table 1) and for NO (Figure 10 and Table 1), not observed by stopped-flow (Figure 3 and Figure 8). 

On the other hand, some difference can be observed between *Hs*-Nb(II) and *Mt*-Nb(II), wherefore the Fe-His bond looks weaker in *Hs*-Nb(II) than in *Mt*-Nb(II), as suggested by EPR spectroscopy for the NO-bound forms, since the Fe-His bond is completely missing already at pH 7.0 in *Hs*-Nb(II)-NO (Figure 2C) while in *Mt*-Nb(II)-NO an equilibrium between a penta-coordinated species and a rhombic one is observed (Figure 2D). Additionally, in the case of the CO-bound form, some difference can be detected between the two Nbs, which shows up in a slightly lower ν(CO) frequency for *Hs*-Nb(II)-CO (i.e., 1950 cm^−1^) with respect to *Mt*-Nb(II)-CO (i.e., 1958 cm^−1^)(Figure 1C), possibly reflecting a different interaction of the ligand with residues of the distal heme pocket. It might be also responsible for the larger geminate recombination, observed after photolysis of *Hs*-Nb(II)-CO (Figure 7), envisaging the possibility of a higher barrier for the ligand escape with respect to *Mt*-Nb(II)-CO, where only a negligible geminate rebinding is observed (Figure 6). No relevant difference instead can be detected between the two Nbs for the Fe-His bound in the CO-bound forms, as indicated by the closely similar fast CO dissociation rate constant (Figure 4) and the similar rate for the relaxation to the equilibrium unliganded conformation after laser photolysis (Figure 6 and Figure 7).

In conclusion, the results here presented show that ferrous Nbs display a significantly reduced reactivity toward exogenous ligands, such as CO and NO, likely due to both (i) H_2_O crowding in the distal side of the heme pocket and (ii) a very high barrier for the concerted movement of the proximal Fe-His bond toward the heme plane upon ligand binding. Such a proximal strain brings about also a severe weakening of the Fe-His proximal bond in the liganded forms, thus leading to a markedly accelerated dissociation rate constants for both CO and NO. Indeed, all the Nb three-dimensional structures determined so far indicate a weakening of the Fe-His-proximal bond, that is 0.10–0.17 Å longer than that observed in *Pc*-Mb [70] and *Ec*-Mb [71]; such bond length differences are meaningful given the high resolution (ranging from 1.79 Å to 1.36 Å) of the three-dimensional structures analyzed. The drastically different regulation of ligand-linked conformational changes in Nbs, as compared to other monomeric hemoproteins (such as mammalian Mbs), is in keeping with the likely different physiological role exerted by this new class of hemoproteins [26].

## 4. Experimental Procedures

### 4.1. Materials

*Mt*-Nb, *At*-Nb, and *Hs*-Nb were cloned, expressed, and purified as described previously [26,29,30]. *Mt*-Nb and *Hs*-Nb concentration was determined spectrophotometrically using the following extinction coefficients at λ_max_ = 407 nm: ε_407 nm_ = 100 mM^−1^ cm^−1^, 80 mM^−1^ cm^−1^, and 147 mM^−1^ cm^−1^, respectively [26].

Gaseous ^12^CO and ^13^CO for Resonance Raman (RR) measurements were purchased from Rivoira (Milan, Italy) and FluoroChem (Hadfield, UK), respectively. Gaseous CO for laser flash photolysis and rapid-mixing stopped-flow kinetics was purchased from Linde AG (Höllriegelskreuth, Germany). The CO solution was prepared by keeping in a closed vessel the 5.0 × 10^−2^ M phosphate buffer solution (pH = 7.0) under CO at *p* = 760.0 mm Hg anaerobically (*T* = 20.0 °C). The solubility of CO in the aqueous buffered solution is 1.03 × 10^−3^ M at *p* = 760.0 mm Hg and *T* = 20.0 °C [51]. NO solutions for UV-Vis and EPR spectroscopy were prepared by dissolving in a phosphate buffer solution (pH = 7.0, *T* = 20.0 °C) sodium dithionite and sodium nitrite (Approx. 1 × 10^‒2^ M). Gaseous NO for rapid-mixing stopped-flow kinetics was purchased from Merck KGA (Darmstadt, Germany). NO was purified by flowing through a glass column packed with NaOH pellets and then by passage through a trapping solution, containing 20 mL of 5.0 M NaOH, to remove traces impurities; the NO pressure was 760.0 mmHg [72]. The NO solution was prepared by keeping in a closed vessel the 5.0 × 10^−2^ M phosphate buffer solution (pH = 7.0) under NO at *p* = 760.0 mm Hg anaerobically (*T* = 20.0 °C). The solubility of NO in the aqueous buffered solution is 2.05 × 10^−3^ M at *p* = 760.0 mm Hg and 20.0 °C [51]. The O_2_ solution for rapid-mixing kinetics was prepared by equilibrating the 5.0 × 10^−2^ M phosphate buffer solution (pH = 7.0) under atmospheric pressure (i.e., *P*_O2_ = 152 mm Hg) at *T* = 20.0 °C. The solubility of O_2_ in the aqueous buffered solution is 1.25 × 10^−3^ M at *p* = 760 mm Hg at *T* = 20.0 °C; therefore, in the air-equilibrated buffer [O_2_] = 2.5 × 10^−4^ M. All the other chemicals were purchased from Merck KGA (Darmstadt, Germany). 

All chemicals were of analytical or reagent grade and were used without further purification unless stated.

### 4.2. Methods

#### 4.2.1. UV-Visible Spectroscopy of Mt-Nb(II), Mt-Nb(II)-CO, Hs-Nb(II), and Hs-Nb(II)-CO

UV-Visible (UV-Vis) spectra of *Mt*-Nb(II) and *Hs*-Nb(II) were collected from 250 to 700 nm using a Cary 300 and a Cary 60 spectrophotometer (Agilent Technologies, Santa Clara, CA, USA) at pH 6.0 (1.0 × 10^−1^ M citrate buffer), 7.4 (2.0 × 10^−2^ M phosphate buffer), and 10.2 (1.0 × 10^−1^ M borate buffer) using a 5 mm nuclear magnetic resonance (NMR) tube (300 nm/min scan rate) or a 1 mm cuvette (600 nm/min scan rate) at 25.0 °C, with a resolution of 1.5 nm. To obtain the complete reduction of *Mt*-Nb and *Hs*-Nb, 2 μL of a freshly prepared 7.8 × 10^−2^ M sodium dithionite solution, previously degassed by flushing with nitrogen, were added to 40 μL of ferric *Mt*-Nb and *Hs*-Nb (*Mt*-Nb(III) and *Hs*-Nb(III), respectively) solutions. The concentration of *Mt*-Nb(II) and *Hs*-Nb(II) ranged between 2.0 × 10^−5^ and 3.0 × 10^−5^ M.

#### 4.2.2. Resonance Raman Measurements of Mt-Nb(II), Mt-Nb(II)-CO, Hs-Nb(II), and Hs-Nb(II)-CO 

*Mt*-Nb(II) and *Hs*-Nb(II) samples were prepared by addition of 2 to 3 μL of a freshly prepared 7.8 × 10^−2^ M sodium dithionite solution to the ferric samples (40 μL) previously degassed by flushing with nitrogen. The *Mt*-Nb(II)-CO and *Hs*-Nb(II)-CO species were prepared by flushing the ferric protein solutions firstly with nitrogen, then with ^12^CO or ^13^CO, and reducing the heme by addition of 2 to 3 μL of a freshly prepared 7.8 × 10^−2^ M sodium dithionite solution. The *Mt*-Nb and *Hs*-Nb concentration ranged between 2.0 × 10^−5^ and 3.0 × 10^−5^ M.

The RR spectra were recorded using a 5 mm NMR tube by excitation with the 413.1 nm line of a Kr^+^ laser (Coherent, Innova 300 °C; Coherent, Santa Clara, CA, USA) and with the 441.6 nm line of a He-Cd laser (Kimmon IK4121R-G; Kimmon Koha Co. LTD, Tokyo, Japan). Backscattered light from a slowly rotating NMR tube was collected and focused into a triple spectrometer (consisting of two Acton Research SpectraPro 2300i and a SpectraPro 2500i in the final stage with a grating of 3600 or 1800 grooves/mm; Princeton Instruments, Trenton, NJ, USA), which works in the subtractive mode, equipped with a liquid nitrogen-cooled CCD detector. Spectral resolution, calculated theoretically based on the optical properties of the spectrometer, was of 1.2 cm^−1^ and spectral dispersion of 0.4 cm^−1^/pixel, and 4 cm^−1^ and spectral dispersion 1.2 cm^−1^ /pixel, for the 3600 and 1800 grating, respectively. This latter grating was used to measure the RR spectra of the CO complexes in the 1800–2300 cm^−1^ region with the 441.6 nm excitation. These spectra were obtained with a cylindrical lens to minimize ligand photolysis since it focuses the laser light into a line instead of a point. The RR spectra were calibrated using as standards carbon tetrachloride, indene, and *n*-pentane, to an accuracy of 1 cm^−1^ for intense isolated bands.

To improve the signal-to-noise ratio, several spectra were accumulated and summed only if no spectral differences were noted. All spectra were baseline corrected. The UV-Vis spectra were measured both prior to and after RR measurements to ensure that no degradation occurred under the experimental conditions that were used. The RR spectra were recorded using the experimental set-up as previously reported [73]. 

#### 4.2.3. UV-Vis Spectroscopy of Mt-Nb(II)-NO and Hs-Nb(II)-NO

UV-Vis spectra of *Mt*-Nb(II)-NO and *Hs*-Nb(II)-NO were collected from 325 to 500 nm using a Cary 8453 spectrophotometer (Agilent Technologies, Santa Clara, CA, USA) at pH 7.0 (5.0 × 10^−2^ M phosphate buffer) employing a 1 cm quartz cuvette at 20.0 °C. *Mt*-Nb(II)-NO and *Hs*-Nb(II)-NO samples were prepared by mixing, inside the cuvette, 0.2 mL of the *Pc*-Mb(III), *Mt*-Nb(III), and *Hs*-Nb(III) solutions (Approx. 5.0 × 10^‒6^ M) with sodium dithionite and sodium nitrite solutions (Approx. 1.0 × 10^‒2^ M).

#### 4.2.4. EPR Spectroscopy of Mt-Nb(II)-NO and Hs-Nb(II)-NO

X-band EPR spectra of *Pc*-Mb(II)-NO*, Mt*-Nb(II)-NO, and *Hs*-Nb(II)-NO were acquired at 110 K and pH 7.0 (3.0 × 10^‒2^ M phosphate buffer). *Pc*-Mb(II)-NO*, Mt*-Nb(II)-NO, and *Hs*-Nb(II)-NO solutions were prepared by mixing, inside the EPR tube, 0.2 mL of the *Pc*-Mb(III), *Mt*-Nb(III), and *Hs*-Nb(III) solutions (Approx. 1.0 × 10^‒4^ M) with sodium dithionite and sodium nitrite solutions (Approx. 1.0 × 10^‒2^ M). Within a few seconds after mixing, the solutions were frozen with liquid N_2_ and the EPR spectrum was recorded [43]. EPR measurements were carried out on a Bruker ELEXSYS E500 spectrometer (Bruker** Bruker BioSpin GmbH, Germany) operating in continuous wave at X band and equipped with a high sensitivity SHQ cavity. A temperature of 110 K was achieved by a nitrogen Bruker VT system. Spectra were recorded using a microwave power of 20 mW and a modulation amplitude of 0.5 mT. Simulation of the EPR spectra was performed with Easypsin v. 5.2.28 [74]. 

#### 4.2.5. CO Binding to Mt-Nb(II) and Hs-Nb(II)

##### Rapid-Mixing Experiments

Kinetics of CO binding to *Mt*-Nb(II) and *Hs*-Nb(II) were investigated spectrophotometrically between pH 2.2 and 7.0 (final buffer concentration, 1.0 × 10^−1^ M phosphate buffer) and 25.0 °C by rapid-mixing ferrous heme-protein solutions (pH 7.0; 1.0 × 10^−3^ M phosphate buffer; final concentration, approx. 2.0 × 10^−6^ M to 5.0 × 10^−6^ M) with CO solutions (pH ≤ 7.0; 2.0 × 10^−1^ M phosphate buffer; final concentration ranging between 2.0 × 10^−5^ M and 2.0 × 10^−4^ M) under anaerobic conditions (i.e., in the presence of 1.0 × 10^−2^ M sodium dithionite) [56]. Kinetics of CO binding to *Mt*-Nb(II) and *Hs*-Nb(II) was recorded over the 380–450 nm wavelength range.

CO binding to *Mt*-Nb(II) and *Hs*-Nb(II) was analyzed in the framework of Scheme 1.

Progress kinetic curves at selected wavelengths were analyzed according to Equation (1): (1)$$OD_{obs}=OD_{0}\pm \sum_{i=1}^{i-n}\Delta OD_{i}e^{\left(-kit\right)}$$
where *OD*_obs_ is the observed optical density at a selected wavelength and at a given time interval, *OD*_0_ is the optical density at *t* = 0, *n* is the number of exponentials, Δ*OD_i_* is the optical density change associated to the exponential *i*, *k_i_* is the pseudo-first-order rate constant of the exponential *i* (i.e., *k*_obs(CO)_) and *t* is the time. Since data collection occurs on a logarithmic scale, experimental points in the first second represent the absolute majority (about 70 out of 100 points) of total collected ones.

Values of the second-order rate constant for *Mt*-Nb(II)-CO and *Hs*-Nb(II)-CO formation (i.e., *k*_on(CO)_) and of the first-order rate constant for CO dissociation from *Mt*-Nb(II)-CO and *Hs*-Nb(II)-CO (i.e., *k*_off(CO)_) were obtained from the dependence of the pseudo-first-order rate constant for *Mt*-Nb(II) and *Hs*-Nb(II) carbonylation (i.e., *k*_obs(CO)_) on the ligand concentration (i.e., [CO]) according to Equation (2):*k*_obs(CO)_ = *k*_on(CO)_ × [CO] + *k*_off(CO)_(2)

The values of the first-order rate constant for CO dissociation from *Mt*-Nb(II)-CO and *Hs*-Nb(II)-CO (i.e., for CO replacement by NO; *k*_off(CO)_) were also determined by mixing the *Mt*-Nb(II)-CO and *Hs*-Nb(II)-CO (final concentration, 2.0 × 10^−6^ M to 5.0 × 10^−6^ M; [CO] = 1.0 × 10^−4^ M) solutions with the NO saturated solution (final concentration, 1.0 × 10^−1^ M), under anaerobic conditions (i.e., in the presence of 2.0 × 10^−3^ M sodium dithionite), at pH 7.0 (1.0 × 10^−1^ M phosphate buffer), and 25.0 °C; no gaseous phase was present [75]. Kinetics of CO dissociation from *Mt*-Nb(II)-CO and *Hs*-Nb(II)-CO was recorded at 421 nm.

The conversion of *Mt*-Nb(II)-CO and *Hs*-Nb(II)-CO to *Mt*-Nb(II)-NO and *Hs*-Nb(II)-NO, respectively, was analyzed in the framework of Scheme 2 [75]:

The values of *k*_off(CO)_ were determined from data analysis, according to Equation (1). The over 100-fold excess of NO over CO guarantees that the reaction proceeds rightward because *k*_on(NO)_ × [NO] >> *k*_on(CO)_ × [CO] [75]. 

The pH-dependence of CO binding kinetics to ferrous Nbs was carried out by mixing in the stopped-flow apparatus the reduced hemoprotein solution (in 1.0 × 10^−3^ M phosphate buffer pH 7.0) with CO solutions at the desired ligand concentration in 0.3 M phosphate and/or acetate buffer titrated to the desired pH value with 1 M NaOH solution. The final pH, reported in Figure 5A,B, was measured immediately on the exit mixture, thus corresponding to the actual pH value in the solution after mixing with CO [56]. Progress kinetic curves were recorded at different wavelengths between 380 and 450 nm and they were analyzed according to Equation (1).

Rapid-mixing experiments were carried out employing an SX18.MV stopped-flow apparatus (Applied Photophysics, Salisbury, UK), equipped with a diode array for spectra acquisition over a 1 ms time range (the light path of the observation chamber was 10 mm) and an SFM-20/MOS-200 rapid-mixing stopped-flow apparatus (BioLogic Science Instruments, Claix, France) (the light path of the observation chamber was 10 mm).

#### 4.2.6. Laser Flash Photolysis Experiments 

CO rebinding kinetics to *Mt*-Nb(II) and *Hs*-Nb(II) at pH 7.4 (5.0 × 10^−2^ M phosphate buffer) was obtained using the second harmonic (532 nm) of a Q-switched Nd:YAG laser (Surelite I-10, Continuum, Santa Clara, CA, USA) and the cw output of a 150 W Xe arc lamp was used as probe beam to monitor absorbance changes at 436 nm. The laser flash photolysis setup was described in detail elsewhere [76].

*Mt*-Nb or *Hs*-Nb (3.0 × 10^−5^ M) were anaerobically reduced with 2 × 10^−3^ M sodium dithionite in sealed 2 × 10 mm quartz cuvette. The CO adducts were obtained by equilibrating solutions in either 0.2 CO atm or 1.0 CO atm. The oxidation state, the molar fraction of ferrous carbonylated Nbs, and the Nb concentration were determined spectrophotometrically. Kinetics were analyzed according to Scheme 3.

According to Scheme 3, there are two types of processes, namely the formation of the A state directly from state B, which is much faster and independent on the ligand concentration (geminate recombination), whose extent *φ_gem_* depends on the relative rates *k**_BA(L)_* and *k**_BC(L)_* according to the following relationship
(3)$$\varphi_{gem}=\;\frac{k_{BA\left(L\right)}}{\left(k_{BA\left(L\right)}+k_{BC\left(L\right)}\right)}$$
and the observed rate *r_gem_* corresponds to
(4)$$r_{gem}=\;k_{BA\left(L\right)}+\;k_{BC\left(L\right)}$$
the formation of state A from state C (through the transient formation of state B), which depends on the ligand concentration (bimolecular recombination) and it is characterized by the second-order rate *r_bim_* corresponding to
(5)$$\;r_{bim}=\;k_{CB\left(L\right)}\;\times \;\varphi_{gem}$$

#### 4.2.7. NO Binding to Mt-Nb(II) and Hs-Nb(II)

##### Rapid Mixing

Kinetics of NO binding to *Mt*-Nb(II) and *Hs*-Nb(II) were investigated spectrophotometrically at pH 7.0 (5.0 × 10^−2^ M phosphate buffer) and 22.0 °C by rapid-mixing ferrous heme-protein solutions (pH = 7.0; 5.0 × 10^−2^ M phosphate buffer; final concentration, 2.0 × 10^−6^ M to 5.0 × 10^−6^ M, respectively) with NO solutions (pH = 7.0; 5.0 × 10^−2^ M phosphate buffer; final concentration ranging between 1.5 × 10^−5^ M and 1.0 × 10^−4^ M) under anaerobic conditions (i.e., in the presence of 1.0 × 10^−2^ M sodium dithionite) [77]. Kinetics of NO binding to *Mt*-Nb(II) and *Hs*-Nb(II) was recorded at 390, 405, and 430 nm.

NO binding to *Mt*-Nb(II) and *Hs*-Nb(II) was analyzed in the framework of Scheme 4.

Progress kinetic curves at selected wavelengths were analyzed according to Equation (1). Values of the second-order rate constant for *Mt*-Nb(II)-NO and *Hs*-Nb(II)-NO formation (i.e., *k*_on(NO)_) were obtained from the dependence of the pseudo-first-order rate constant for *Mt*-Nb(II) and *Hs*-Nb(II) nitrosylation (i.e., *k*_obs(NO)_) on the ligand concentration (i.e., [NO]) according to Equation (6):*k*_obs(NO)_ = *k*_on(NO)_ × [NO](6)

Rapid-mixing experiments were carried out employing an SFM-20/MOS-200 rapid-mixing stopped-flow apparatus (BioLogic Science Instruments, Claix, France) (the light path of the observation chamber was 10 mm).

#### 4.2.8. Laser Flash Photolysis Experiments

NO rebinding kinetics were studied with the same laser flash photolysis setup used in the CO rebinding experiments, using 405 nm as the observation wavelength.

*Hs*-Nb(II) was prepared by anaerobically adding a fresh solution of sodium dithionite (1 mM final concentration) to a *Hs*-Nb(III) solution previously equilibrated with N_2_. An anaerobically prepared 2 mM solution of MAHMA NONOate was then added to reach a final concentration of 100 µM (NO equivalents). The concentrations of the *Hs*-Nb solutions ranged between 2.0 × 10^−5^ and 3.0 × 10^−5^ M. The ligation forms were checked by absorption spectrophotometry using a Cary 4000 spectrophotometer (Agilent Technologies, CA, USA). No measurements are reported for *Mt*-Nb(II)-NO because of the instability of this form over the relatively long time intervals of the experiment (about 1 h). 

#### 4.2.9. NO Dissociation from Mt-Nb(II)-NO and Hs-Nb(II)-NO

Values of the first-order rate constant for NO dissociation from *Mt*-Nb(II)-NO and *Hs*-Nb(II)-NO (i.e., for NO replacement by CO; *k*_off(NO)_) were determined by mixing the freshly prepared (within 5 min after their preparation) *Mt*-Nb(II)-NO and *Hs*-Nb(II)-NO (final concentration, 2.0 × 10^−6^ M to 5.0 × 10^−6^ M, respectively; [NO] = 5.0 × 10^−6^ M) solutions with the CO saturated solution (final concentration, 5.0 × 10^−4^ M), under anaerobic conditions (i.e., in the presence of 2.0 × 10^−3^ M sodium dithionite), at pH 7.0 (5.0 × 10^−2^ M phosphate buffer), and 22.0 °C; no gaseous phase was present [53]. Kinetics of CO dissociation from *Mt*-Nb(II)-CO and *Hs*-Nb(II)-CO was recorded at 420 nm. 

The conversion of *Mt*-Nb(II)-NO and *Hs*-Nb(II)-NO to *Mt*-Nb(II)-CO and *Hs*-Nb(II)-CO, respectively, was analyzed in the framework of Scheme 5.

The values of *k*_off(NO)_ were determined from data analysis, according to Equation (1). The over 100-fold excess of CO over NO guarantees that the reaction proceeds rightward because *k*_on(CO)_ × [CO] > *k*_on(NO)_ × [NO] (Scheme 2). 

Kinetics of NO dissociation from *Mt*-Nb(II)-NO and *Hs*-Nb(II)-NO was carried out employing an SFM-20/MOS-200 rapid-mixing stopped-flow apparatus (BioLogic Science Instruments, Claix, France); the light path of the observation chamber was 10 mm. 

#### 4.2.10. O_2_ Dissociation from Mt-Nb(II)-O_2_, At-Nb(II)-O_2_, and Hs-Nb(II)-O_2_

Kinetics of O_2_ dissociation from the transient *Mt*-Nb(II)-O_2_, *At*-Nb(II)-O_2_ and *Hs*-Nb(II)-O_2_ species were investigated spectrophotometrically at pH 7.0 (1.0 × 10^−1^ M phosphate buffer) and 25.0 °C by oxygen pulse experiments [75]. This type of reaction proceeds according to Scheme 6.

The deoxygenation of *Mt*-Nb(II)-O_2_ and *Hs*-Nb(II)-O_2_ process was followed at 431 nm. Kinetics of *Mt*-Nb(II)-O_2_ and *Hs*-Nb(II)-O_2_ deoxygenation were analyzed according to Equation (1).

Rapid-mixing experiments were carried out employing an SX18.MV stopped-flow apparatus (Applied Photophysics, Salisbury, UK) equipped with a diode array for spectra acquisition over a 1 ms time range; the light path of the observation chamber was 10 mm. 

### 4.3. Data Analysis

Spectroscopic data were analyzed using LabCalc (Galactic Industries Corporation, Salem, NH, USA) and OriginPro (OriginLab Corporation, Northampton, MA, USA). Kinetic data were analyzed using the MatLab (The Math Works Inc., Natick, MA, USA), the OriginPro (OriginLab, Northampton, MA, USA), and the GraphPad Prism (GraphPad Software, La Jolla, CA, USA) programs. The results are given as mean values of at least four experiments plus or minus the corresponding standard deviation.

## Data Availability

Not applicable.

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
