# Peer review of "Mycobacterial and Human Ferrous Nitrobindins: Spectroscopic and Reactivity Properties"

_ijms, 2021, doi:10.3390/ijms22041674_

Round 1

Reviewer 1 Report

The authors should make more explicit in their introduction how these studies are different from earlier results. 

The who did what statement should be explain at a detailed level ie per each method used. 

Furthermore I need a copy of their submission letter explaining how this paper reports results that are new compared to their earlier works. The statement of who did what is too generic and should instead be detailed as to who measured the various types of spectra.

Author Response

Reviewer 1

Reviewer 1.1. The authors should make more explicit in their introduction how these studies are different from earlier results.

Author 1.1. The last paragraph of the Introduction has been revised as follows (page 2, lines 14-31): “This paper reports a deep investigation of spectroscopic and functional properties of ferrous Mycobacterium tuberculosis and Homo sapiens Nbs (Mt-Nb(II) and Hs-Nb(II), respectively). For the first time: (i) UV-Vis, RR, and EPR spectroscopic properties of ligand-free and ligand-bound Mt-Nb(II) and Hs-Nb(II) have been recorded; (ii) kinetics of CO, NO and O2 binding to Mt-Nb(II) and Hs-Nb(II) has been investigated by rapid-mixing stopped-flow technique and laser-flash photolysis; and (iii) spectroscopic and functional data have been analyzed in parallel with those of Arabidopsis thaliana Nb(II) [29]. Present data indicate that upon ligand binding, the Fe(II) atom of Nbs moves onto the heme plane, this brings about a marked lengthening of the proximal Fe-imidazole bond leading to its rupture. This structural evidence is accompanied by a marked enhancement of ligand dissociation rate constants. Moreover, these data highlight the conservation of Nbs in bacteria, plants and animals, and indicate that structural-functional relationships in Nbs strongly differ from those of prototypical mammalian myoglobins, such as Equus caballus (Ec-Mb) and/or Physeter catodon (Pc-Mb), and of Rhodnius prolixus nitrophorins (Rp-NPs). This suggests that Nbs play a functional role clearly distinct from other eukaryotic and prokaryotic heme-proteins.”

Reviewer.1.2. Furthermore, I need a copy of their submission letter explaining how this paper reports results that are new compared to their earlier works.

Authors 1.2. According to the Reviewer’s request, here follows a copy of the letter sent to the Editor following the request to clarify the novelty of this manuscript compared to the earliest works.

“Nitrobindins (Nbs) are monomeric all-β-barrel heme-proteins that have been found from bacteria to Homo sapiens; in contrast, all-β-barrel nitrophorins that have been found only in Rhodnius prolixus and Cimex lectularius. Therefore, Nbs may be taken as a molecular molecular model of all-β-barrel heme-proteins.

At present, only ferrous Arabidopsis thaliana Nb and ferric Mycobacterium tuberculosis and Homo sapiens Nbs have been investigated from the structural and functional viewpoints.

This paper reports a deep investigation of spectroscopic and functional properties of ferrous Mycobacterium tuberculosis and Homo sapiens Nbs highlighting the conservation of Nbs in bacteria, plants and animals.

For the first time:

(1) UV-Vis and Resonance Raman spectroscopic properties of ligand-free and ligand-bound ferrous Mycobacterium tuberculosis and Homo sapiens Nbs have been deeply investigated highlighting the weakness of the proximal His-Fe bond. Of note nor UV-Vis neither Resonance Raman spectroscopic properties are available for ferrous Arabidopsis thaliana Nb.

(2) UV-Vis and EPR spectroscopic properties of ferrous nitrosylated Mycobacterium tuberculosis and Homo sapiens Nbs have been investigated highlighting the cleavage of the proximal His-Fe bond. The penta-coordination of the metal center at neutral pH is very unusual.

(3) Kinetics of CO and NO binding to ferrous Mycobacterium tuberculosis and Homo sapiens Nbs have been investigated by both rapid-mixing stopped flow and laser photolysis. Of note, kinetics of CO and NO binding to ferrous Arabidopsis thaliana Nb have been obtained previously only by laser photolysis. Therefore, both steady-state and transient functional properties have been analyzed firstly.

(4) Kinetics of O2 dissociation from ferrous oxygenated Mycobacterium tuberculosis and Homo sapiens Nbs have been investigated by rapid-mixing stopped flow. Since no data were available from literature for Arabidopsis thaliana Nb, kinetics of O2 dissociation have been here determined for the purpose of a homogeneous comparison.

Present data indicate that upon ligand binding, the Fe(II) atom of Nbs moves onto the heme plane, this brings about a marked lengthening of the proximal Fe-imidazole bond leading to its rupture. This structural evidence is accompanied by a marked enhancement of ligand dissociation rate constants. Moreover, these data clearly indicate that structural-functional relationships in Nbs strongly differ from what observed in mammalian and truncated hemoproteins, suggesting that Nbs play a functional role clearly distinct from other eukaryotic and prokaryotic hemoproteins."

Reviewer.1.3. The statement of who did what is too generic and should instead be detailed as to who measured the various types of spectra.

Authors 1.3. The following detailed description of Author Contributions is reported in the revised manuscript (page 27):

AUTHORS CONTRIBUTION: G.D.S., A.d.M., P.A., L.T., G.S., UV-Vis and RR spectroscopy; S.D.M., P.F., D.P., EPR spectroscopy; G.D.S., A.d.M., P.A., C.C., M.C. ligand biding kinetics by rapid-mixing stopped-flow technique; C.V., S.A., S.B. ligand biding kinetics by laser flash photolysis; G.D.S., A.d.M., A.P., C.C., L.T., S.A., S.B., S.D.M., P.F., D.P., M.B. helped in the planning the manuscript and experiments; M.B., G.S., C.V., M.C., P.A. designed the manuscript and experiments; M.B., G.S., C.V., M.C., P.A., G.D.S., C.C., S.D.M., P.F., D.P. wrote the manuscript.

Reviewer 2 Report

The authors studied several nitrobindins proteins from several organisms. A lot of experiments have been done in anaerobic conditions to have information about the rate constants of binding of NO or CO to the heme dependent proteins. Introduction is quite short, and a lot of results from literature are cited during the results section to compare the results of the authors with the results available in the literature and concerning proteins from others organisms. A better separation of the Introduction / Results and Discussion sections could be interesting to present more clearly what it has already done, the new results and finally the conclusion of the study.

Comments:

-There is a problem in all the schemes concerning the protein ligand interaction. All arrows are missing and the rate constants are shifted form the reaction. It is also difficult to check and understand the message.

-The Table 3 is missing also.

-The equation 6 is cited at least 2 times but I don’t found it neither in the text nor in the methods description.

- Did the authors check the results obtain by stopped-flow by doubling the concentration of their proteins? It could be good to see that type of results to check that the observed signal is due to the protein and not from other modifications.

- For Figure 3A, could it be possible to add a graph with all the original traces that gave the exponential curves and that gave the kobs. In Figure 3B, what is the concentration of CO that give these traces. What means relative traces? Relative to what? At which wavelength has been measured the absorbance? Why the authors didn’t give the original traces with the absorbance? Could the authors add the residual traces between the experimental and fitted traces. How many points are there registered during the 1 first second of the trace? This information is not found in the experimental part of the paper.

- Figure 4A. What is it in ordonate? kobs? It is write Iobs? How the authors managed the variation of pH? In experimental section, they indicate a phosphate buffer. But the buffering effect can not be achieved among this large range with phosphate only. And the authors certainly add salts to adjust the pH, so, the ionic strength is certainly not constant among the range of pH used. Could the authors detailed a little bit more the preparation of the several buffers to study the pH effects?

- Figure 7 and more generally for all the experiments presented in the paper: how many shots have been done for the stopped-flow experiments (or for laser flash or other experiments presented in the paper). How many independent experiments? How many shots are averaged? The authors mention that the error bar is smaller than the point but statistical information are missing (number of shot, number of batch of proteins etc…). Figure 7C is obtained from which kobs determined from which wavelength?

- Figure 8: Where are the experimental points? Under the line obtained from equation 1? Could the authors showed the experimental results and the fit on the same graph? (same comment for Figure 7A, 7B and Figure 10)

Author Response

Reviewer 2

Reviewer 2.1. Introduction is quite short, and a lot of results from literature are cited during the results section to compare the results of the authors with the results available in the literature and concerning proteins from others organisms. A better separation of the Introduction / Results and Discussion sections could be interesting to present more clearly what it has already done, the new results and finally the conclusion of the study.

Author 2.1. We have followed advises from the reviewer, significantly enriching the last paragraph of Introduction (page 2, lines 14-31) of the revised version.

Reviewer 2.2. There is a problem in all the schemes concerning the protein ligand interaction. All arrows are missing and the rate constants are shifted form the reaction. It is also difficult to check and understand the message.

Author 2.2. For the sake of clarity, Schemes have been reported as pictures that have been pasted in the revised manuscript.

Reviewer 2.3. The Table 3 is missing also.

Author 2.3. Table 3 has been mentioned erroneously. Data mentioned in the text refer to Table 1. The typing error has been corrected (caption of Figure 3). Tables 1 and 2 have been reformatted and pasted in the revised manuscript.

Reviewer 2.3. The equation 6 is cited at least 2 times but I don’t found it neither in the text nor in the methods description.

Author 2.3. Eqn. 6 has been reported erroneously. Data analysis refers to eqn. 1. The typing error has been corrected (captions of Figure 3 and 10). Eqn. 7, reported erroneously in the caption of Figure 3, has been changed to eqn. 2. Eqn 4 (page 25, line 17) has been also corrected.

Reviewer 2.4. Did the authors check the results obtain by stopped-flow by doubling the concentration of their proteins? It could be good to see that type of results to check that the observed signal is due to the protein and not from other modifications.

Author 2.4. Under pseudo-first-order conditions (i.e., [ligand] > 5 × [Nb]), values of kobs are independent of the protein concentration. This point has been addressed in the text (page 7, lines 35-36; page 15, lines 21-22; page 22, line 35; page 23, line 19; page 25, lines 1-2 and 37). For the sake of clarity, values of rate constants have been reported in Table 1 of Supplementary Materials.

Reviewer 2.5. For Figure 3A, could it be possible to add a graph with all the original traces that gave the exponential curves and that gave the kobs. In Figure 3B, what is the concentration of CO that give these traces.

Author 2.5. New panels (A and B) showing the original traces of CO binding to ferrous Nbs have been reported in the revised manuscript. The caption of Figure 3 has been amended accordingly. The time-course of CO dissociation from Mt-Nb(II)-CO and Hs-Nb(II)-CO have been reported in the new Figure 4.

Reviewer 2.6. What means relative traces? Relative to what? At which wavelength has been measured the absorbance? Why the authors didn’t give the original traces with the absorbance? Could the authors add the residual traces between the experimental and fitted traces. How many points are there registered during the 1 first second of the trace? This information is not found in the experimental part of the paper.

Author 2.6. Figures 3, 8, 9, and 11 have been redrawn and analyzed as suggested by the Reviewer. The figure captions have been revised accordingly. It appears evident that, since data collection has been carried out on a logarithmic scale, about 60-70% of data points have been collected within the first second.

Reviewer 2.7. Figure 4. What is it in ordinate? kobs? It is write Iobs? How the authors managed the variation of pH? In experimental section, they indicate a phosphate buffer. But the buffering effect can not be achieved among this large range with phosphate only. And the authors certainly add salts to adjust the pH, so, the ionic strength is certainly not constant among the range of pH used. Could the authors detail a little bit more the preparation of the several buffers to study the pH effects?

Author 2.7. In Figure 5 of the revised version (corresponding to Fig. 4 of the previous version) lobs has been changed to kobs. The pH-jump method has been described in the Methods section of the revised manuscript  (page 23, lines 37-44). The relevant literature has been reported (Refs 43; 48; 49; 50, 56, 57; 58; 59; 60; 61; 62).

Reviewer 2.8. Figure 7 and more generally for all the experiments presented in the paper: how many shots have been done for the stopped-flow experiments (or for laser flash or other experiments presented in the paper). How many independent experiments? How many shots are averaged? The authors mention that the error bar is smaller than the point but statistical information is missing (number of shot, number of batch of proteins, etc…). Figure 7C is obtained from which kobs determined from which wavelength?

Author 2.8. For all the experiments presented in the paper, the details requested from the Reviewer are reported in the “4.3. Data Analysis” section. In the stopped-flow and laser-photolysis experiments, at least four shots have been done for each experimental condition. At least four independent experiments have been done for ligand binding to and dissociation from ferrous (ligated-)Nbs. Six batches of each Nb have been used for spectroscopic and functional investigations.

Reviewer 2.9. Figure 7C is obtained from which kobs determined from which wavelength?

Authors 2.9. kobs(NO) values. shown in Figure 7C, are the average of those obtained at 390 nm, 405 nm, and 430 nm. The caption of Figure 7C has been amended accordingly.

Reviewer 2.8. Figure 8: Where are the experimental points? Under the line obtained from equation 1? Could the authors show the experimental results and the fit on the same graph? (same comment for Figure 7A, 7B, and Figure 10)

Author 2.8. Figure 8 has been redrawn as suggested by the Reviewer.

Round 2

Reviewer 2 Report

The authors improved the introduction, the quality of results presentation and the methods. They just did'nt mentionned the number of points for the stopped-flow and laser flash experiments. Concerning the exponential curves analysis, they wrote that kobs is independent of the protein concentrations, that's good, but they did'nt check for the amplitude of the exponential. Did the authors do the experiments with various concentration of proteins?

Author Response

Reviewer 1. The authors just did not mentioned the number of points for the stopped-flow and laser flash experiments.

Authors 1. Actually, in the first reply letter we mentioned that 60-70% of the experimental points were acquired within the first second, since the data collection was on a logarithmic scale. However, in the second round revision we added this point in the description of the kinetic data collection (page 23, lines 7-9 of the revised manuscript).

Reviewer 2. Concerning the exponential curves analysis, they wrote that kobs is independent of the protein concentrations, that is good, but they did not check for the amplitude of the exponential.

Authors 2. The amplitude of the exponentials is dependent on the CO concentration under all the experimental condition, since CO concentration is similar to the value of the dissociation equilibrium constant K(CO) (i.e., [CO] does not fully saturate Mt-Nb(II) and Hs-Nb(II)) (see page 7, lines 34-37 of the revised manuscript).

The amplitude of the exponentials is independent of the NO concentration under all the experimental conditions since the NO concentration was larger by at least three orders of magnitude than the dissociation equilibrium constant K(NO) (i.e., [NO] was largely sufficient to saturate both Mt-Nb(II) and Hs-Nb(II)) (see page 15, lines 21-24 of the revised manuscript).

Reviewer 3. Did the authors do the experiments with various concentration of proteins?

Authors 3. As already reported, kinetic experiments were conducted with protein concentrations ranging between 1.0×10‒6 to 5.0×-10‒6 as stated in the revised manuscript at page 22 (line 34), page 23 (line 20), page 25 (lines 1-2 and line 37), page 26 (line 22). Kinetics were unaffected from the protein concentration (page 7, lines 34-35, page 15, lines 21-22).

Round 3

Reviewer 2 Report

The authors answered to my questions. They explained the variation of amplitude between the two ligands used in the study. It seems a solid paper. The authors did a lot of modifications in the text and it improved and made a difference between the first submission.